# Long-lived spin waves in a metallic antiferromagnet

G. Poelchen [1,2,3] ✉, J. Hellwig[4], M. Peters[4], D. Yu. Usachov [5], K. Kliemt [4], C. Laubschat[2], P. M. Echenique[5,6], E. V. Chulkov[5,7,8], C. Krellner [4], S. S. P. Parkin [9], D. V. Vyalikh [5,6], A. Ernst [9,10] & K. Kummer [1] ✉

Collective spin excitations in magnetically ordered crystals, called magnons or spin waves, can serve as carriers in novel spintronic devices with ultralow energy consumption. The generation of well-detectable spin flows requires long lifetimes of high-frequency magnons. In general, the lifetime of spin waves in a metal is substantially reduced due to a strong coupling of magnons to the Stoner continuum. This makes metals unattractive for use as components for magnonic devices. Here, we present the metallic antiferromagnet $CeCo_2P_2$, which exhibits long-living magnons even in the terahertz (THz) regime. For $CeCo_2P_2$, our first-principle calculations predict a suppression of low-energy spin-flip Stoner excitations, which is verified by resonant inelastic X-ray scattering measurements. By comparison to the isostructural compound $LaCo_2P_2$, we show how small structural changes can dramatically alter the electronic structure around the Fermi level leading to the classical picture of the strongly damped magnons intrinsic to metallic systems. Our results not only demonstrate that long-lived magnons in the THz regime can exist in bulk metallic systems, but they also open a path for an efficient search for metallic magnetic systems in which undamped THz magnons can be excited.

Magnons or spin waves are elementary quasiparticles, which represent a collective motion of magnetic moments in ordered systems. Spin waves can propagate in materials and therewith transport a spin current[1–4]. This spin flow requires no electrical charge transport and therefore no electrical losses creating Joule heating. Spin waves enclose a wide frequency range, from gigahertz up to a few hundreds of terahertz[5]. The larger the excitation frequency, the faster are the magnon processes. However, the utilisation of magnons in magnonic applications can be limited by their lifetime, which can range from a few tenths of a microsecond down to tens of femtoseconds, depending on their frequency and some features of the electronic structure[6]. Low-frequency (GHz) magnons usually posses longer lifetimes than THz magnons and are therefore currently considered as the most promising for magnonic applications[3,7,8]. But one of the exciting prospects of magnonics is clearly the potential to eventually be able to design ultrafast magnonic devices pushing far into the THz regime[9]. To this end, materials with high ordering temperatures and high magnon energies, and at the same time long magnon lifetimes, i.e. weak magnon damping, are required.

[1]European Synchrotron Radiation Facility, 71 Avenue des Martyrs, 38043 Grenoble, France. [2]Institut für Festkörper- und Materialphysik, Technische Universität Dresden, 01062 Dresden, Germany. [3]Max Planck Institute for Chemical Physics of Solids, Nöthnitzer Straße 40, 01187 Dresden, Germany. [4]Kristall- und Materiallabor, Physikalisches Institut, Goethe-Universität Frankfurt, Max-von-Laue Strasse 1, 60438 Frankfurt am Main, Germany. [5]Donostia International Physics Center (DIPC), 20018 Donostia-San Sebastián, Spain. [6]IKERBASQUE, Basque Foundation for Science, 48011 Bilbao, Spain. [7]Centro de Física de Materiales (CFM-MPC), Centro Mixto CSIC-UPV/EHU, 20018 Donostia-San Sebastián, Spain. [8]Departamento de Polímeros y Materiales Avanzados: Física, Química y Tecnología, Facultad de Ciencias Químicas, Universidad del País Vasco UPV/EHU, 20080 Donostia-San Sebastián, Spain. [9]Max-Planck-Institut für Mikrostrukturphysik, Weinberg 2, 06120 Halle, Germany. [10]Institut für Theoretische Physik, Johannes Kepler Universität, 4040 Linz, Austria. ✉e-mail: georg.poelchen@esrf.fr; kurt.kummer@esrf.fr

Giga- and terahertz magnons can occur in magnetic insulators. Due to the absence of conducting electrons, magnon damping in these materials can be weak and charge flow (electric current) can be naturally excluded. However, the absence of conducting electrons also suppresses indirect magnetic exchange which can negatively affect ordering temperatures. As a consequence, there is only a limited number of magnetic insulators suitable for magnonic applications[8,10]. Itinerant (metallic) magnets typically possess both high magnon frequencies and high critical temperatures because conduction electrons promote indirect exchange interaction. However, these positive aspects are countered by the strong damping that magnons experience in metallic systems. The resulting short magnon lifetimes are a major obstacle for the use of itinerant magnets in terahertz magnonic devices[11–13].

Magnon lifetimes can be affected by various scattering processes such as electron-magnon, phonon-magnon, magnon-magnon interactions, spin-orbit damping or impurity scattering[9,13]. In metals, the most significant is the electron-magnon interaction – the interaction between magnons and the Stoner continuum, which represents single-particle excitations in the same energy range as the magnons[11,14–16]. For this reason, THz magnons usually have a short lifetime in many magnetic materials and their use in magnonics is strongly restricted[11]. The Stoner excitations are governed by transitions from occupied to unoccupied states without or with spin flips. When low-energy Stoner excitations are not allowed, magnons can exhibit long lifetimes up to high frequencies[17,18]. This can happen if occupied and unoccupied electronic states participating in Stoner excitations, are separated by a gap (Stoner gap). The size of the Stoner gap should be larger than the range of the magnon frequencies. While in halfmetals and insulators, this can occur naturally due to the presence of an electronic (spin) gap, a Stoner gap can also exist in magnetic metals, when conducting states, the states at the Fermi level e.g. $sp$ bands, do not contribute to the Stoner excitations. In insulators, the density of states (DOS) at the Fermi level is zero for both spin channels, but, as it was mentioned above, the absence of conductance electrons reduces strongly the interaction between magnetic moments and, as a consequence, the ordering temperature and magnon frequencies. Halfmetallicity, i.e. a vanishing DOS near the Fermi level in only one spin channel, is predicted by theory, for instance, in some oxides, Heusler alloys or topological insulators[17]. However, in experimental realisations, the effect of halfmetallicity can easily be spoiled by various defects[19]. Therefore, a search of materials, in which the magnon damping can be controlled and substantially reduced, remains to be an important challenge in magnonics. The discovery of a low-damping material with high-frequency magnons would push the progress in the field of terahertz magnonics one big step further.

Magnons can be generated in many types of magnetic systems: ferro-, ferri- and antiferromagnetic materials. The later have a great potential in spintronics and magnonics applications[20]. The key features of antiferromagnets such as ultrafast magnetisation dynamics, small magnetic susceptibility, negligible net magnetisation are considered to be fruitful in the emerging field of antiferromagnetic spintronics. One big advantage of antiferromagnetic materials is their robustness against various perturbations such as ultrafast dynamic or parasitic stray fields. Because of the antiparallel exchange coupling antiferromagnets can exhibit much faster spin dynamics as conventional ferromagnets, which application field is limited mainly in the GHz range due to their inherent slower magnetisation dynamics. The exchange interaction in antiferromagnetic systems is usually very strong and, therefore, intrinsic natural frequencies of the spin wave modes are in the THz range. In addition, antiferromagnets have naturally more than one magnetic sublattices and, therefore, multiple dynamic modes can be generally observed. This makes antiferromagnets, both insulators and metals, extremely well suited for spin transport and domain wall motion[21,22].

However, a direct observation of the antiferromagnetic magnon modes for nonzero wave vectors is one of the most important challenge in magnonics. Short wavelength and high-frequency magnon modes inherent to the antiferromagnetic dynamics make those measurements quite difficult. Therefore magnetic damping measurements and direct observation of antiferromagnetic magnon modes are crucial future challenges. Because of their great potential for the next generation of spintronic applications[20,23] the search of antiferromagnetic materials with long lifetime, high-frequency magnons in the THz range is one of the ambitious goals of the modern condensed matter physics.

Due to a large Stoner gap, spin waves in antiferromagnetic insulators and halfmetals have very low damping. This attractive property is however spoiled by the fact that the magnetisation in these materials is hardly detectable and, as result, a fast read of switching signals is not possible. The invisible nature of magnetisation and high resonant frequencies hinder experimental investigation and detection of antiferromagnetic magnons, restricting the measurements only on the localised dynamics at zero wave vector. Nevertheless, there is a class of antiferromagnets, which do not suffer on the above discussed weaknesses. These are synthetic antiferromagnets, which consists of ferromagnetic layers separated by a non-magnetic spacer[24]. The ferromagnetic layers interact weakly antiferromagnetically via the Ruderman-Kittel-Kasuya-Yoshida (RKKY) or magnetic dipole interactions. Spin waves in these materials are usually well-detectable and one can also observe a clear cross-over between the ferromagnetic and antiferromagnetic dynamics. However, the antiferromagnetic coupling in most synthetic antiferromagnets is usually small and the induced magnons are in the GHz range. But finally, if the problem of the short magnon lifetimes could be neglected, directly using metallic systems with their large magnon frequencies would perhaps be the most attractive option from a practical point due to being generally well suited for nano-fabrication techniques as well for integration with existing high-frequency technology[1] while being more power efficient[25,26].

Hence, we take a closer look at the lanthanide intermetallic antiferromagnets with LnT$_2$X$_2$ (Ln is a lanthanide element, T a transition metal element and X a main group element) which can experience 4$f$, 3$d$ or a cross-over between 4$f$ and 3$d$ spin dynamics[27–29]. We show both experimentally and theoretically that one member of this family, the metallic antiferromagnet CeCo$_2$P$_2$, which has a very high ordering temperature $T_N$ = 440 K, possesses both a large Stoner gap and large magnon frequencies. As a consequence, undamped magnons up to the THz range can be generated in this material, with all the advantages for magnonic devices discussed above. This makes CeCo$_2$P$_2$ a promising new basis for magnonic quantum devices within the THz regime.

To recapitulate the above discussion, a promising material for metallic THz magnonics should fulfil several criteria. It should exhibit a large ordering temperature and large magnetic moments, which are both signs of a strong magnetic coupling and a stable magnetic phase necessary for spin excitations in the THz regime. To strongly suppress spin-flip Stoner excitations, the magnetically-active states (Stoner states) should be gapped around the Fermi level $E_F$, which would result in a reduced total DOS at $E_F$ composed only of $sp$ states responsible for the metallic character. Figure 1a, b schematically shows how the continuum of Stoner excitations shifts towards higher energies as the spin gap around $E_F$ increases. Stoner states cause a significant damping of magnons if the magnon excitations hybridise strongly with the Stoner continuum. In metals, spin wave excitations can typically reach energies > 250 meV, i.e. THz frequencies. When Stoner states are located high enough in energy, the electron-magnon interaction, i.e. the hybridisation between the spin wave and the Stoner states, is not significant and magnons can posses long lifetimes up to very high frequencies. In order to achieve that, the occupied and unoccupied electronic states which participate in Stoner excitations between the

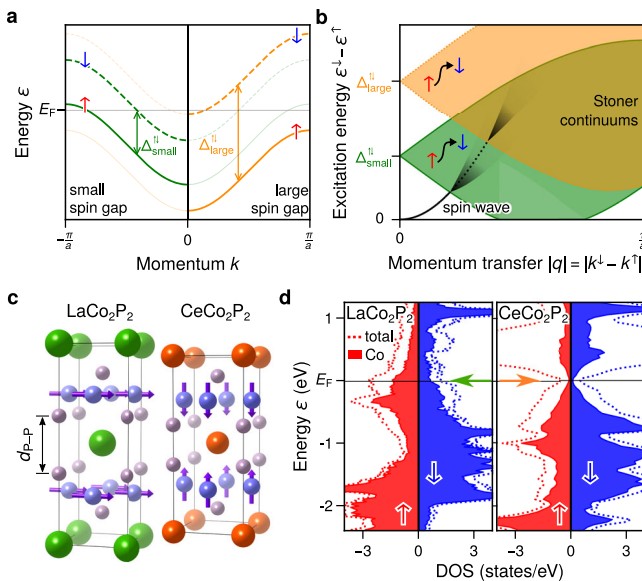

**Fig. 1 | Correlation between spin gap and spin wave scattering. a** Metallic band structure of a spin-split band for a small and large spin (Stoner) gap. **b** Relationship between the corresponding Stoner continuum and the onset of the spin wave scattering. Larger spin gaps promote undamped spin waves up to higher energies. **c** Crystal structure of $LaCo_2P_2$ and $CeCo_2P_2$ with the reduced distance and larger magnetic exchange between Co layers in $CeCo_2P_2$. **d** Spin-resolved density of states in $LaCo_2P_2$ and $CeCo_2P_2$.

two spin channels should be separated by a Stoner gap or pseudogap that is larger than the energy range of the spin wave excitations. Thus large Stoner gaps are typical for magnetic semiconductors and insulators or in halfmetals, in which there is a energy gap between the electronic states, but are not expected for magnetic metals where the Stoner states usually contribute to the Fermi surface.

We found that the intermetallic system $CeCo_2P_2$[28] is a rare exception from this general behaviour. In $CeCo_2P_2$, the Co moments have a stacked antiferromagnetic (AFM) order along the $c$ axis with each layer being ferromagnetically (FM) ordered along $c$ below $T_N \approx 440$ K[30]. The extraordinarily high ordering temperature and the large Co moments of $0.94\,\mu_B$ in comparison to similar systems make this material very attractive for practical applications. Most importantly, first-principle calculations for this material suggest a reduced density of states around $E_F$[28,31], while a recent high-throughput calculations even propose $CeCo_2P_2$ to be an enforced semi-metal with topological properties[32]. Therefore, $CeCo_2P_2$ seems to be a promising candidate to exhibit long-living THz magnons.

In contrast to most Ce intermetallics, there are no Ce $4f$ contributions to the specific heat which can be interpreted as Ce behaving tetravalently[28,30]. The tetravalent behaviour is also evidenced by a reduced unit cell volume leading to a short distance between the phosphorus atoms $d_{P-P}$ and thus to a strongly three-dimensional character.

For comparison, we choose the isostructural $LaCo_2P_2$, which differs from $CeCo_2P_2$ by only one electron per formula unit, but both the structural as well as the magnetic properties differ strongly. As a result of the lanthanide contraction going from La to Ce, the $c$ lattice constant is much larger for $LaCo_2P_2$ leading to a larger $d_{P-P}$. Hence, $LaCo_2P_2$ is in the so-called uncollapsed phase with weaker interlayer couplings and a more two-dimensional character. This becomes apparent in the different magnetic order with the Co moments ordering ferromagnetically in the $ab$ plane. Both, the ordering temperature $T_C \approx 135$ K and the Co moments $\mu_{Co} \approx 0.45\,\mu_B$ are significantly lower than in $CeCo_2P_2$[33–35]. In addition, de Haas-van Alphen measurements[36] as well as a large Sommerfeld coefficient[28] indicate a

large DOS at $E_F$. We would therefore expect a much weaker magnon dispersion and a strong electron-magnon damping.

## Results and discussion

Our density-functional-theory calculations (DFT) reveal these differences between $CeCo_2P_2$ and $LaCo_2P_2$ in the electronic densities of states, as depicted in Fig. 1d. In $CeCo_2P_2$, the total DOS is reduced at $E_F$ and the magnetically-active occupied and unoccupied Co states are well separated by a pseudogap. In $LaCo_2P_2$ there is a large total DOS with strong contribution from $3d$ states at $E_F$ in both spin channels which is typical for a magnetic metal. Thus, for both compounds, we expect metallic characters due to the nonzero total DOS at $E_F$, but the electron-magnon interaction in $CeCo_2P_2$ should be much weaker than in $LaCo_2P_2$ or other metallic systems. In addition, because of the lanthanide contraction effect, the exchange interaction in $CeCo_2P_2$ is anticipated to be larger than in $LaCo_2P_2$, which should be reflected in higher energies of the magnon excitations.

A direct way to confirm our expectations is to use resonant inelastic X-ray scattering (RIXS) at the Co $L_3$ edge to directly measure the spin excitations and their damping over a large **q** and energy $\hbar\omega_q$ interval. In Fig. 2a, the geometry of the RIXS process is presented. Incoming light $\mathbf{k}_{in}$ under the incidence angle $\theta$ is scattered of the sample $\mathbf{k}_{out}$ under the scattering angle $2\theta'$ resulting in a momentum transfer $\mathbf{q} = \mathbf{k}_{in} - \mathbf{k}_{out}$. Tuning the incoming photon energy to the Co $L_3$ absorption edge allows for the element specific enhancement while changing the scattering geometry allows measuring the elementary excitations in the sample as a function of momentum transfer **q**. In Fig. 2c, we show the experimentally observed RIXS intensity for $CeCo_2P_2$ as a function of energy loss $\hbar\omega_{in} - \hbar\omega_{out}$ at a fixed $\mathbf{q} = (-0.12, 0, 1)$. The measured spectrum is dominated by two features, a double peak structure around zero energy loss and a broad peak around 1 eV to 3 eV. Slightly tuning the incident photon energy, the peaks in the low energy region will remain at the same energy loss as expected for Raman-like quasi-particle excitations. By contrast, the broad feature centred around 1.5 eV will move on the energy loss axis with incident photon energies which identifies it as stemming from fluorescence decay of the core hole. This very intense fluorescence feature due to local, intraatomic decay of the RIXS intermediate state is a characteristic of RIXS spectra taken from metallic systems with high electron mobility[37–39].

Here we will concentrate on the low energy region of the RIXS spectra shown in Fig. 2d with the quasielastic peak at zero energy loss and an additional peak which varies in energy position as a function of **q** and can be attributed to magnon excitations. The dispersion along the $(H\,0\,1)$ direction with $H$ going from −0.24 to 0.2 (r.l.u.), i.e. the full $H$ range accessible for the Co $L_3$ edge in this compound, is shown in Fig. 2e. The dispersion seems to follow a linear relation as expected for antiferromagnets. Due to the magnetic Bragg condition, measurements close to $\mathbf{q} = (0, 0, 1)$ are dominated by the elastic peak and omitted here.

In order to determine both the energy position $\hbar\omega_q$ and the damping of the magnetic excitations, we fitted the spectra using a model with two peaks, namely an elastic peak and a magnon peak, and a background function as shown in Fig. 2d. The elastic line is described by a single line at zero energy loss, while the magnon peak is generally described by function based on the damped harmonic oscillator model

$$I(E_{loss} > 0, \hbar\omega_q, \gamma) = I_0(\hbar\omega_q) \frac{\gamma E_{loss}}{\left(E_{loss}^2 - \hbar^2\omega_q^2\right)^2 + 4\gamma^2 E_{loss}^2} \quad (1)$$

with a damping factor $\gamma$[40–43]. Both peaks are convolved with a Gaussian with FWHM $\Delta E = 28$ meV to account for the experimental resolution in our experiment. This two peak model gives an excellent description of

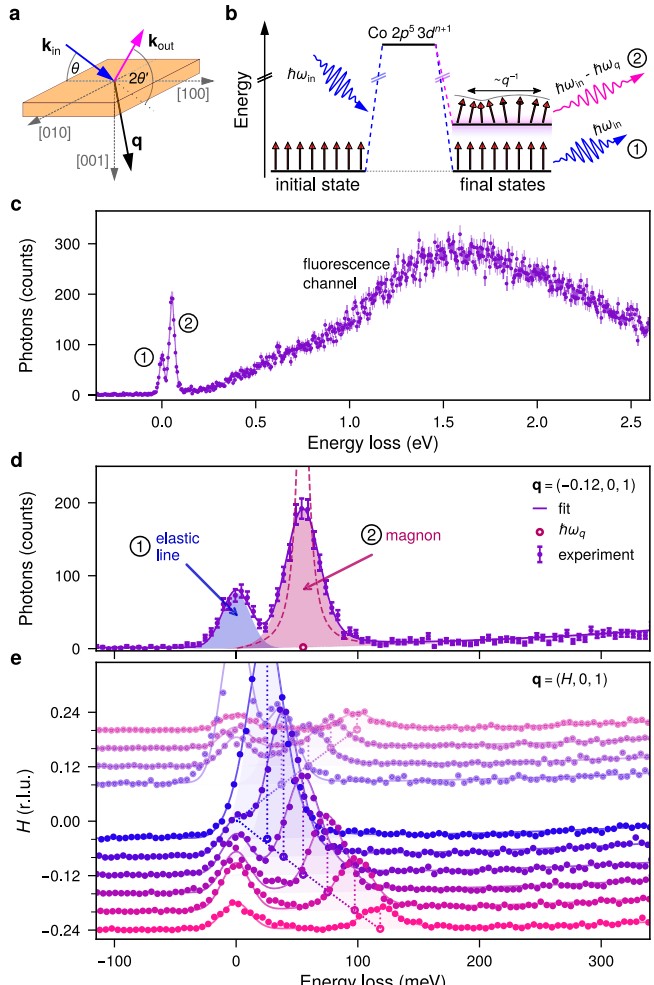

**Fig. 2 | RIXS measurements of the magnon excitations in CeCo$_2$P$_2$.**
**a** Experimental geometry. **b** Schematic of the RIXS process, showing elastic scattering associated with (1) no energy loss and (2) magnon excitations at finite energy loss. **c** Experimentally observed RIXS spectrum at $\mathbf{q} = (-0.12, 0, 1)$. **d** Zoom into the low energy region decomposed into the (1) quasielastic line around zero energy loss and (2) a sharp loss feature due to magnon excitations. The fit of the experimental data is plotted above the data as continuous line. The dashed line shows the magnon peak before convolution with the experimental resolution.
**e** $\mathbf{q}$ dependence of the RIXS spectra. The linear dispersion of the magnon peak is highlighted as dotted line as guide to the eye.

the experimental data. Some small residual intensity can be seen for larger $H$ values around 30 to 40 meV, which corresponds to the typical excitation energies of phonons in this class of systems[44]. Since these contributions are very small, their influence will be neglected.

In order to describe the magnetic excitations in CeCo$_2$P$_2$ theoretically, we determined the exchange coupling constants $J_i$ applying a magnetic force theorem as it is implemented within a first-principles multiple scattering theory[45,46]. We found that the interlayer coupling $J_i$ dominates leading to AFM coupling (negative $J_3$) between neighbouring Co layers and ferromagnetic coupling within the Co layers, in agreement with what is observed experimentally. The calculated magnon dispersion, rescaled by a factor 0.9, is shown in Fig. 3a together with the experimentally values determined with RIXS (symbols) for the high-symmetry directions.

We find good overall agreement between experiment and theory with a linear dispersion $\omega_q \propto q$ for $q_{\parallel} \to 0$ as expected for AFM order[47]. Along the $(H\,0\,1)$ and $(H\,H\,1)$ directions the spin wave energies extend beyond 400 meV, i.e well into the THz regime. Along the $\Gamma-Z$ direction, the dispersion is much smaller. We note that while the shape of the

dispersions compares nicely, the calculations slightly overestimated the spin waves. Applying a constrained fit with a scaled theoretical dispersion showed that the best agreement with the experiment is achieved when rescaling the theoretical dispersion of 0.9. The slight disagreement between experiment and theory can be attributed to strong electronic correlation effects, which cannot be fully taken into account in DFT calculations. However, the overall agreement with the experimental results is excellent.

Besides the magnon dispersion, the RIXS measurements make it possible to determine the damping factor $\gamma$ which we introduced in Equation (1). In Fig. 3a, the damping factor $\gamma$ is shown as error bar. Since $\gamma \ll \hbar\omega_q$, the damping is minimal and the magnon is in the undamped regime. In this case, the shape of the experimentally observed magnon peak is nearly fully symmetric and its width determined primarily by the experimental resolution ($2\gamma < \Delta E$). For a detailed discussion of how even a small damping $\gamma$ affects the RIXS spectra, see Supplementary Information, section 2. The experimental results are in a good agreement with lifetimes calculated within a first-principles linear response theory for the magnetic susceptibility[17,48]. Theory also provides lifetimes for higher magnon frequencies which are not accessible in experiments and shown as inverse lifetime interval around the calculated dispersion. We can see that in the calculations notable damping, $\gamma \geq 10$ meV, sets in only for very high magnon energies above 200 meV. In order to confirm these theoretical results, we also acquired RIXS spectra away from the $(H\,0\,1)$ high-symmetry direction by fixing the scattering angle to the maximum possible 150° and moving along the path $\mathbf{q} = (-0.24, 0, 1) \to (-0.5, 0, 0.45)$. In this way, high enough in-plane momentum transfers can be achieved to be able to probe the spin wave excitations up to 200 meV and above. In agreement with theory, we find that notable damping only sets in at very high magnon energies of 200 meV and above.

For comparison, Fig. 3b shows the theoretically determined magnon dispersion for LaCo$_2$P$_2$ along the high-symmetry directions scaled again by a factor of 0.9. Clearly visible is the much weaker dispersion in comparison to CeCo$_2$P$_2$, a result of a reduced magnetic coupling and smaller magnetic moments. The first interlayer coupling constant $J_3$ was found to be close to zero in accordance with the large $d_{P-P}$ distance and experimentally observed ferromagnetic ordering. The second obvious difference is the large inverse lifetime which is in the order of excitation energy of the magnon.

The experimental comparison of the RIXS spectra for CeCo$_2$P$_2$ and LaCo$_2$P$_2$ for similar magnon excitation energies and for the same $\mathbf{q}$ point is shown in Fig. 4a. Comparing first the CeCo$_2$P$_2$ spectra measured at $\mathbf{q} = (\pm 0.08, 0, 1)$ and the LaCo$_2$P$_2$ spectra at $\mathbf{q} = (-0.24, 0, 1)$, where the magnon excitations have the same energy of about $\hbar\omega_q = 34$ meV, the difference is striking. In CeCo$_2$P$_2$ the magnon excitations appear as a well separated, symmetric peak with a close to resolution-limited width. In LaCo$_2$P$_2$, by contrast, the magnon excitations give rise to a broad asymmetric peak merging with the quasielastic line with a width primarily defined by the lifetime (damping) of the magnons. Comparing the spectra for both compounds at the same $\mathbf{q} = (-0.24, 0, 1)$, the difference in the magnon dispersion is clearly visible. For CeCo$_2$P$_2$, the magnon excitation energy is around 120 meV and thus nearly three times larger than for LaCo$_2$P$_2$. Even at this magnon excitation energy, the determined damping factor $\gamma$ is still smaller than the energy resolution and the magnon peak shape is mostly symmetric and primarily governed by resolution broadening. Additional temperature-dependent measurements at $\mathbf{q} = (-0.24, 0, 1)$ show that the properties of the excited magnons in CeCo$_2$P$_2$ are not notably altered up to room temperature, due to the very high ordering temperature $T_N = 440$ K of the material (see Supplementary Information, section 3).

These results are in accordance with the calculated inverse lifetimes as well as our predictions based on the exceptional density of states of CeCo$_2$P$_2$ resulting in a large effective Stoner gap. We note that

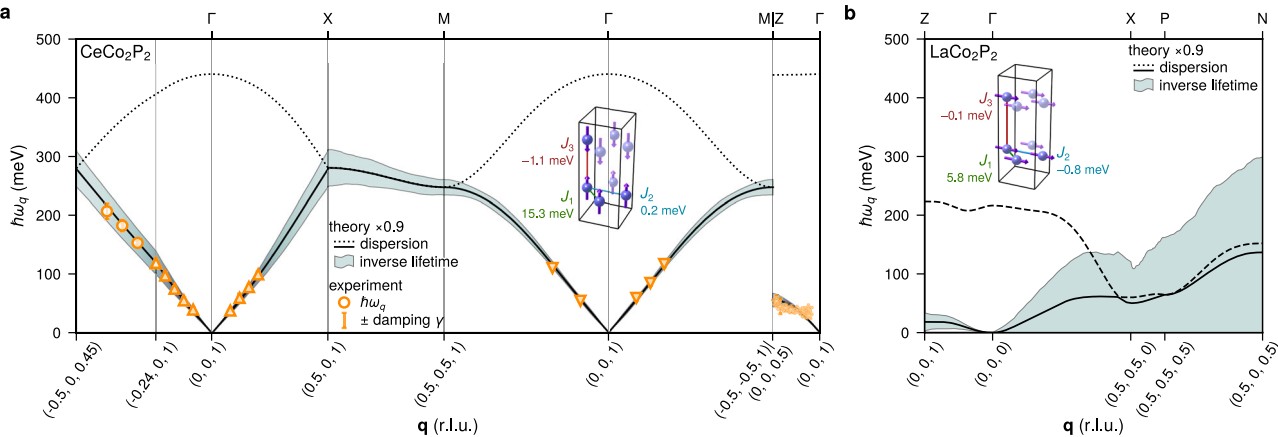

**Fig. 3 | Experimental and theoretical magnon dispersion and lifetime.**
**a** Calculated and experimentally observed magnon dispersion and inverse lifetimes in antiferromagnetic $CeCo_2P_2$. The dashed lines show backfolded magnon branches with vanishing RIXS cross section. The calculated inverse lifetimes correspond to the solid lines. The inset shows the first three $J_i$ of the Heisenberg Hamiltonian. **b** The corresponding theoretical results for ferromagnetic $LaCo_2P_2$.

recently the effect of a narrow Stoner continuum on spin wave excitations has been discussed[49]. Such a narrow Stoner continuum would arise from very flat, spin polarised bands around $E_F$. In such a scenario Stoner spin flip excitations would occur only in a very narrow energy band and long-lived spin-waves could possibly exist at energies above and below. Our electronic structure calculations (Fig. 1d) as well as recent ARPES data[28] give no indication of flat bands in the bulk of $CeCo_2P_2$. The very low Sommerfeld coefficient found in specific heat measurements[28] further excludes the presence of heavy quasiparticles close to $E_F$. Therefore, we believe that a scenario based on a very localised Stoner continuum cannot be invoked for $CeCo_2P_2$.

Our findings are summarised in Fig. 4b where we show the experimental and theoretically obtained magnon damping with respect to the magnon excitation energy for $CeCo_2P_2$. For comparison we also added our results for $LaCo_2P_2$ as well as magnon dampings reported in the literature for other metallic magnets[16,39,40,50,51]. Clearly, the properties of $CeCo_2P_2$ with its high-energy, long-lifetime magnons and at the same time intermetallic behaviour stand-out in this comparison. In order to quantitatively compare the magnon damping in $CeCo_2P_2$ and other systems, we can look at the damping ratio for the magnetic excitations, i.e. the ratio between the damping $\gamma$ and the magnon energy $\hbar\omega_q$. The inverse of the damping ratio is a measure for the longevity of the spin excitations. $\hbar\omega_q/(2\gamma) \leq 1$ corresponds to an overdamped system were the high energy spin excitations get scattered immediately after creation, while for $\hbar\omega_q/(2\gamma) \gg 1$ the spin excitations are long-lived even for high magnon energies. In Fig. 4c, we show this inverse damping ratio for various metallic magnets versus the highest experimentally observed magnon excitation energy. The exceptional properties of $CeCo_2P_2$ compared to other metallic magnets become immediately visible with its very long magnon lifetimes, i.e. weak electron-magnon interaction, and its large magnon energies.

*In conclusion*, on the example of the stoichiometric room temperature antiferromagnet $CeCo_2P_2$, we demonstrate the generation of long-living high energy magnons in a metallic system. Such materials are potentially very attractive for magnonic applications with ultralow energy consumption, with high detectability and with high-speed magnonic processes. Our calculations indicate that the long-living magnetic excitations are a result of a weak electron-magnon interaction due to a strongly reduced density of Co 3d states around the Fermi level resulting in an opening of the Stoner gap. For the isostructural compound $LaCo_2P_2$ which has a large DOS at the $E_F$ we observed strongly damped magnetic excitations already well below 100 meV, which is the typical response of magnetic metals.

We also note that the elaborated methodology can be applied for the computational search of metallic magnetically ordered materials where similar or more advanced magnetic excitations from 3d and 4f sublattices can be generated. Other members of the Co pnictide family, but also some Fe pnictides may be a good starting point for such a search. Especially materials where significant enhancements in the ordering temperature can be induced as a function of external or internal stimulus appear very promising, as these changes are usually related to strong modifications of the electronic structure of the magnetically active states around $E_F$. In the case of $LaCo_2P_2$, progressive replacement of La ions in $LaCo_2P_2$ with Ce eventually causes a volume collapse that strongly enhances magnetic interactions and the magnetic ordering temperature, and opens a gap in the d density of states[31]. Both together, the strongly enhanced magnetic exchange and the gap in the d DOS at $E_F$ are important for the observation of long-lived, high-energy magnons in $CeCo_2P_2$, while the sp DOS at $E_F$ preserves good metallicity of the material. Like in the case of $LaCo_2P_2$ and $CeCo_2P_2$, we note that Fe pnictides can show very high energy spin waves with very different degrees of spin wave damping and that, for instance, the damping in $SrFe_2As_2$ or $BaFe_2As_2$ is much more pronounced than in $CaFe_2As_2$. This hints to a high degree of tunability of magnon damping in these materials, too. A similar volume collapse like in $CeCo_2P_2$ together with a notable enhancement in ordering temperature was also seen for $CaFe_2As_2$ under application of pressure of a few GPa[52]. Chemical tuning or substrate induced strain may be promising routes towards an electronic structure at $E_F$ qualitatively similar to that in $CeCo_2P_2$ and thus strongly reduced magnon damping also in this system. Materials like $CeCo_2P_2$ could also be used as building blocks in engineered thin film structures. In this way, they can be combined with other magnetically ordered layers to engineer novel nanostructures with 3d and 4f long-lived magnons in the THz regime which opens great opportunities for the development of novel functionalities in magnonic applications.

# Methods
## Calculations
Electronic structure calculations were performed using a first-principles Green function method[46] within the density functional theory in a generalised gradient approximation[53]. Localised Co 3d states were treated with a GGA+U approximation to take into account strong electronic correlations[54,55]. The value of the Hubbard $U$ parameter was chosen by comparing the calculated Néel

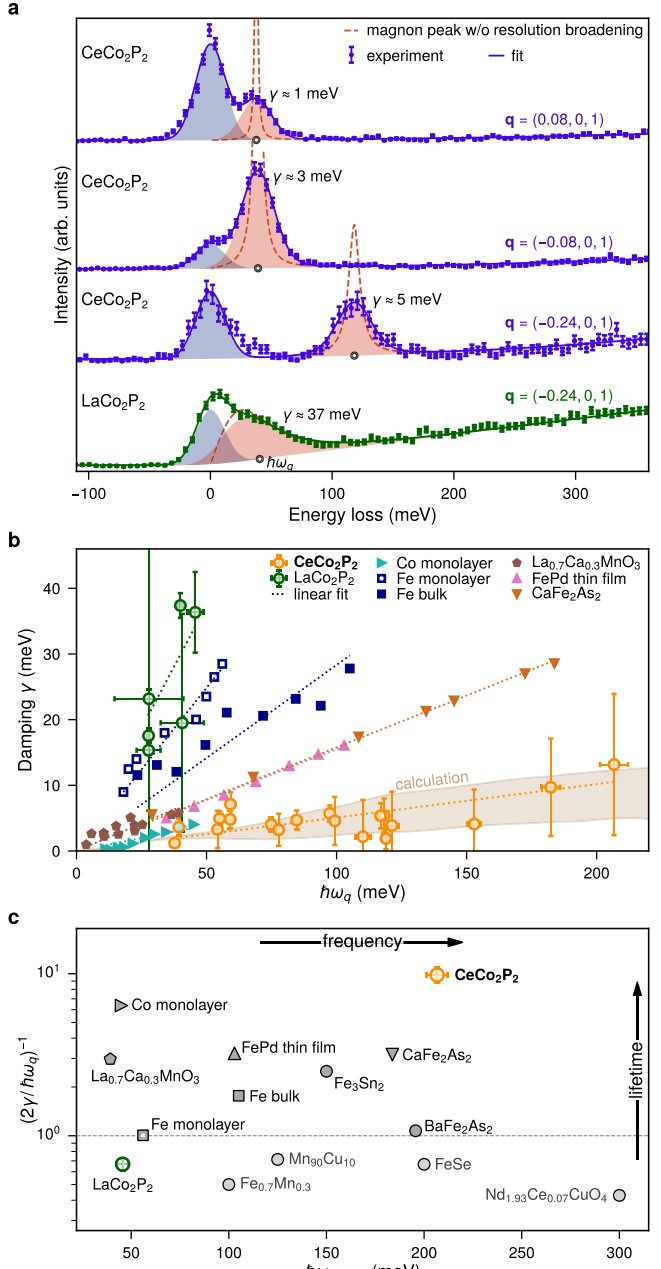

**Fig. 4 | Comparison of magnon lifetimes and frequencies in metallic magnets.**
**a** RIXS spectra of $CeCo_2P_2$ and $LaCo_2P_2$ for different **q** vectors. $CeCo_2P_2$ shows sharp, resolution-limited peak shapes in contrast to a broad magnon peak for $LaCo_2P_2$. The shaded areas correspond to the elastic and magnon peak after resolution broadening. The magnon dispersion is considerable smaller in $LaCo_2P_2$ in agreement with calculation. **b** Experimental and calculated damping of the magnetic excitations in $CeCo_2P_2$ and $LaCo_2P_2$ compared to data reported for other metallic magnets[16,39,40,50,51]. **c** Inverse magnon damping ratio vs highest experimentally observed magnon excitation energy $\hbar\omega_{q,\,max}$ of $CeCo_2P_2$ compared to other metallic magnets[16,38–40,50,51,59–65].

temperature with the experimental one. The Néel temperature was evaluated using the Heisenberg model within the random phase approximation. The best agreement was achieved for the Hubbard parameter $U = 3$ eV ($T_{N,theo} = 420$ K, $T_{N,exp} = 440$ K). The exchange parameters were calculated using a magnetic force theorem as it is implemented within the multiple scattering theory[45,46]. Magnonic lifetimes were estimated using a simplified approach based on a linear response theory for magnetic susceptibility[17,48].

## RIXS

RIXS measurements were performed at the ID32 beamline of the ESRF[56]. The combined energy resolution during the experiment was determined to be $\Delta E = 28$ meV FWHM using the low resolution, high efficiency gratings in the beamline and RIXS spectrometer. All presented spectra were obtained from high-quality single crystals[57] freshly cleaved before the RIXS measurements, although measurements on an uncleaved sample gave the same RIXS spectra, confirming the bulk origin of the magnetic excitations. The spectra shown in the manuscript were acquired at $T = 20$ K, with the energy set to slightly above the white line of the Co $L_3$ absorption edge, $h\nu \approx 780$ eV. Additional temperature-dependent measurements up to room temperature are reported in the Supplementary Information file.

## Data availability

All data that support the findings of this study are shown in the paper and the Supplementary Information. They are also available from the ESRF Data Portal under the following https://doi.org/10.15151/ESRF-DC-1307993516[58].

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

## Acknowledgements

This work was supported by the German Research Foundation (DFG) through Grants No. KR3831/5-1, No. LA655/20-1m, Fermi-NEst, GRK1621, TRR288 (No. 422213477) project A03, and SFB1143 (No. 247310070). D.V.V. acknowledges support from the Spanish Ministry of Science and Innovation, project PID2020-116093RB-C44, funded by MCIN/AEI/10.13039/ 501100011033. A.E. acknowledges funding by Fonds zur Förderung der wissenschaftlichen Forschung (FWF) grant I 5384. Part of the calculations were performed at Rechenzentrum Garching of the Max Planck Society (Germany). The authors acknowledge the ESRF for beamtime on beamline ID32.

## Author contributions

K.Ku. conceptualised the research; J.H., M.P., K.Kl. and C.K. grew and characterised the samples; G.P. and K.Ku. performed the RIXS experiment and experimental data analysis; A.E. performed the theoretical modelling; the obtained experimental and theoretical results were discussed together with D.V.V., K.Kl., C.K., S.S.P.P. and E.V.C.; the original draft was written by G.P. and K.Ku. with edits from C.K., A.E. and D.V.V.

## Competing interests

The authors declare no competing interests.
