## [Peer Review File · Nature Communications]

Reviewers' Comments:

Reviewer #1:

Remarks to the Author:

The paper proposed that an antiferromagnetic metal, CeCo₂P₂, can be taken as a promising magnonic material for its unusual low spin wave damping. The authors claimed that DFT calculations suggest a very low DOS (or a larger Stoner gap) at Fermi level in CeCo₂P₂, which results in a low magnon damping due to the suppression of Stoner excitations. They further employed RIXS to carefully measure the magnonic dispersion and observe the magnons with a long lifetime. The study is meaningful to the field of magnonics and may be important for practical applications. The experimental data are collected and analyzed in a high-quality way. The conclusion is attractive and reasonable. Basically I think the paper deserves to appear in Nat. Comm. I still have some technical questions before I could recommend its publication.

-- CeCo₂P₂ has a low DOS at Fermi level which seems a key reason for its low spin wave damping. It means that CeCo₂P₂ is a "bad" metal. In principle this is not a very surprising conclusion, though the material realization is challenging. Is it possible to generalize the conclusion and look for the compounds with a low magnon damping in so called "bad" metals? The authors may put some discussions on this.

-- Related to the above question, the low DOS at Fermi level should give rise to a reduced indirect exchange coupling and hence cause a lower ordering temperature, according to the explanations in the introduction. But the AFM temperature in CeCo₂P₂ is much higher than that in LaCo₂P₂ which has a much higher DOS at Fermi level. How to understand the possible inconsistency?

-- It is curious how trivalent rare earth ion (La) in LaCo₂P₂ evolves into tetravalent ion (Ce) in CeCo₂P₂. Is there any further evidence for the tetravalent Ce?

-- The strong electron correlation in 3d ions has been taken into account using LDA + U method. Normally such kind of calculations need to be corrected by experimental electronic structures. Otherwise the calculated band structures may seriously deviate from the experimental ones, like in cuprates. Did the authors compare their calculations with any experimental results?

Reviewer #3:

Remarks to the Author:

The main objective of this research is to investigate the spin wave dispersion in CeCo₂P₂ and LaCo₂P₂. Through RIXS measurements, the authors observed long-living magnons with high energy in the metallic antiferromagnet CeCo₂P₂. Their calculations suggest that the low damping of the magnons is due to a weak magnon-electron interaction resulting from a reduced density of state around the Fermi level in CeCo₂P₂. The authors claimed that, compared to other itinerant magnets, CeCo₂P₂ is potentially attractive for spintronic applications with ultralow energy consumption.

However, in my opinion, the evidence presented by the authors is not convincing enough to prove the undamped spin waves. The damping factor extracted from the RIXS measurement is even smaller than the energy resolution, which raises some doubts. To support their argument, the authors need more powerful evidence, such as inelastic neutron scattering measurement with higher resolution. If we accept the authors' claim that the inelastic peak in the RIXS spectra is a magnetic excitation, then the lack of significant damping below 200 meV could be attributed to the low instrumental resolution. Additionally, it is unclear how small the damping factor can be defined as undamped spin waves. For instance, the parent compound iron-based superconductor CaFe₂As₂, which has a damping factor of around 30 meV at 200 meV that is comparable to current CeCo₂P₂, is still considered a damped spin wave system (Nature Physics 5, 555–560 (2009)). Therefore, I cannot recommend that the current version of the manuscript be published in Nature Communications.

1. As authors mentioned that their CeCo₂P₂ is a metallic antiferromagnet. Are they able to provide more experimental evidences to demonstrate that it exhibits metallic properties.
2. Authors mentioned "This very intense fluorescence feature due to local, intraatomic decay of the

RIXS intermediate state is a characteristic of RIXS spectra taken from metallic systems with high electron mobility" in the page 8. I'm interested in finding out whether this can differentiate the surface state of CeCo₂P₂, given that there is a theoretical computation suggesting that it is a topological semi-metal.

3. The authors referred to CeCo₂P₂ as an itinerant antiferromagnet in the title. Whether authors try to state the origin of magnetism in CeCo₂P₂ arises from quasiparticle excitations of a nested Fermi surface? If not, this title may be misleading to readers.

4. Have temperature-dependent measurements been conducted to verify that the inelastic peaks in the RIXS spectra are magnetic excitations?

5. I attempted to extract the damping factor from the RIXS spectra of CeCo₂P₂ using the function described in the main text. However, the value I obtained was approximately 14 meV at 60 meV, which is almost twice the value shown in the data plot in Fig. 4b. To enable readers to replicate the analysis, the authors should provide a more explicit explanation of how they extracted the damping factor.

6. The authors should take great care to review the manuscript for spelling errors. Mistakes such as "overestimateded", "calulcated", "interant", and so on should be avoided.

Reply to Reviewer #1

Reviewer 1:

The paper proposed that an antiferromagnetic metal, CeCo₂P₂, can be taken as a promising magnonic material for its unusual low spin wave damping. The authors claimed that DFT calculations suggest a very low DOS (or a larger Stoner gap) at Fermi level in CeCo₂P₂, which results in a low magnon damping due to the suppression of Stoner excitations. They further employed RIXS to carefully measure the magnonic dispersion and observe the magnons with a long lifetime. The study is meaningful to the field of magnonics and may be important for practical applications. The experimental data are collected and analyzed in a high-quality way. The conclusion is attractive and reasonable. Basically I think the paper deserves to appear in Nat. Comm. I still have some technical questions before I could recommend its publication.

Authors:

We would like to thank the reviewer for the critical reading of our manuscript and are pleased to hear that the reviewer thinks it deserves publication in Nature Comm. Below we reply to the reviewer's questions which have helped greatly in further improving our manuscript.

Reviewer 1:

CeCo₂P₂ has a low DOS at Fermi level which seems a key reason for its low spin wave damping. It means that CeCo₂P₂ is a "bad" metal. In principle this is not a very surprising conclusion, though the material realization is challenging. Is it possible to generalize the conclusion and look for the compounds with a low magnon damping in so called "bad" metals? The authors may put some discussions on this.

Authors:

The referee is right that as one goes from metallic to insulating behaviour with local moment magnetism one would expect to find less and less damped magnons. For instance, in insulating AFM ordered parent compounds of cuprate superconductors largely undamped spin waves up to high energies are present. However, this behaviour gets lost as soon as these systems are doped into the bad metal phase where very strong magnon damping is seen. We therefore do not believe that our findings can easily be generalised to bad metals. Moreover, from the experimentally observed resistivity of CeCo₂P₂ we would also not necessarily classify CeCo₂P₂ as a bad metal either (data is now shown as a Supplementary Information, please also see our reply to question 1 of Ref. 3). To us the important ingredient is the quasi bandgap in the *d* electron DOS while the sizable *sp* density at E_F allows to maintain good electrical conductivity. We are thankful of the referee for bringing up this important point and we rephrased the corresponding sentences in the manuscript to highlight explicitly the importance of the suppression of the Co states at E_F , which are the main contributor to the spin-flip transitions.

Reviewer 1:

Related to the above question, the low DOS at Fermi level should give rise to a reduced indirect exchange coupling and hence cause a lower ordering temperature, according to the explanations in the introduction. But the AFM temperature in CeCo₂P₂ is much higher than that in LaCo₂P₂ which has a much higher DOS at Fermi level. How to understand the possible inconsistency?

Authors:

The magnetic interaction in solids can be of various origins. In the introduction, we have mentioned conduction electrons, which can actively mediate indirect interaction between magnetic moments. In many systems this mechanism can be very significant and our calculations show that this is also the case for CeCo₂P₂. The strength of this interaction depends on the size of the magnetic moments, on the distance between the moments and on the weight of spin polarised conduction electrons, which can be roughly estimated as the DOS of quasi-free states at the Fermi level. The Co magnetic moment is notably smaller in LaCo₂P₂ than in CeCo₂P₂ (0.6 μ_B vs. 0.9 μ_B). At the same time, the distance between the Co layers in the LaCo₂P₂ is substantially larger than in CeCo₂P₂. By contrast, the density of quasi-free *sp* electron states actively participating in the indirect exchange interaction is very similar in both systems. For these reasons, the exchange interaction in the CeCo₂P₂ is significantly stronger than that in the LaCo₂P₂, giving rise to a significantly higher magnetic ordering temperature and larger magnon energy dispersion.

Note that while the presence of *d* states at the Fermi level has little effect on the mediation of indirect exchange it is the underlying cause for the strong magnon damping in LaCo₂P₂, and magnetic metals in general. These states form the Stoner continuum related to spin flip transitions that strongly damping excited spin waves.

Reviewer 1:

It is curious how trivalent rare earth ion (La) in LaCo₂P₂ evolves into tetravalent ion (Ce) in CeCo₂P₂. Is there any further evidence for the tetravalent Ce?

Authors:

The bulk properties of CeCo₂P₂ have been thoroughly studied and unanimously established the unusual tetravalent behaviour of the Ce ions [R1-R5]. Strong evidence for that comes from the very small Sommerfeld coefficient, even smaller compared to LaCo₂P₂, where no *4f* contributions are present. From this thermodynamic data it is clear that no localised *4f* electrons are present in CeCo₂P₂. This is shown in Figure R1 below, which was recently published in [R5]. In addition, the cell volume in CeCo₂P₂ is considerably smaller than what would be expected from the normal lanthanide contraction, if every lanthanide is in the trivalent state. This is shown in the right panel of Figure R1 where we plot the unit cell volumes reported in Reehuis and Jeitschko, Phys. Chem. Solids 51, 961 (1990). The red band represents the expected volume, if Ce would be trivalent in CeCo₂P₂. The observed volume is considerably smaller, which can be only explained by strong deviations from trivalent behaviour. However, such a behaviour is not unconventional and observed in other intermetallic systems with tetravalent Ce, where the related La-based compound has trivalent La. Further evidence against a trivalent valence state comes from the absence of any signs of local Ce moments in neutron diffraction

or magnetisation measurements [Reehuis et al., Journal of Alloys and Compounds 266, 54 (1998)]. For a more detailed discussion, we can refer the referee to a recent ARPES study where we discussed the tetravalent behaviour of Ce in the bulk in detail [R5].

Figure R1: (a) The Sommerfeld coefficients γ of both LaCo_2P_2 and CeCo_2P_2 , which were determined from the heat-capacity measurements $C(T)$ between 1.8 and 10 K [R5]. (b) Unit cell volume of the LnCo_2P_2 series. The expected lanthanide contraction for trivalent lanthanide ions is shown in red.

Reviewer 1:

The strong electron correlation in 3d ions has been taken into account using LDA + U method. Normally such kind of calculations need to be corrected by experimental electronic structures. Otherwise the calculated band structures may seriously deviate from the experimental ones, like in cuprates. Did the authors compare their calculations with any experimental results?

Authors:

In our calculations we used the Neel temperature of the system to fix the Hubbard U parameter. The Neel temperature is a macroscopic observable, which can be calculated using the Heisenberg model with exchange coupling constants obtained from quantum mechanical calculations. The exchange constants are directly related to the electronic and magnetic structure. Therefore, a good agreement between theory and experiment for the Neel temperature is indicative of a good choice for the Hubbard U parameter. The Neel temperature was estimated within a random phase approximation, which slightly underestimates the critical temperature. Our result, 420 K, obtained with $U=3 \text{ eV}$ agrees well with the experiment ($T=440 \text{ K}$).

Reply to Reviewer #3

Authors:

We would like to thank the Reviewer for reading our manuscript and the critical comments. They helped us improving our manuscript. We have also added a Supplementary Information document that contains further data and information related to the referee's comments. Before we reply to the specific questions we would like to address the general concern raised by the referee.

Reviewer 3:

The main objective of this research is to investigate the spin wave dispersion in CeCo₂P₂ and LaCo₂P₂. Through RIXS measurements, the authors observed long-living magnons with high energy in the metallic antiferromagnet CeCo₂P₂. Their calculations suggest that the low damping of the magnons is due to a weak magnon-electron interaction resulting from a reduced density of state around the Fermi level in CeCo₂P₂. The authors claimed that, compared to other itinerant magnets, CeCo₂P₂ is potentially attractive for spintronic applications with ultralow energy consumption.

However, in my opinion, the evidence presented by the authors is not convincing enough to prove the undamped spin waves. The damping factor extracted from the RIXS measurement is even smaller than the energy resolution, which raises some doubts. To support their argument, the authors need more powerful evidence, such as inelastic neutron scattering measurement with higher resolution. If we accept the authors' claim that the inelastic peak in the RIXS spectra is a magnetic excitation, then the lack of significant damping below 200 meV could be attributed to the low instrumental resolution. Additionally, it is unclear how small the damping factor can be defined as undamped spin waves. For instance, the parent compound iron-based superconductor CaFe₂As₂, which has a damping factor of around 30 meV at 200 meV that is comparable to current CeCo₂P₂, is still considered a damped spin wave system (Nature Physics 5, 555–560 (2009)). Therefore, I cannot recommend that the current version of the manuscript be published in Nature Communications.

Authors:

Firstly, the reviewer suggests that our RIXS results should be substantiated with INS as the limited experimental resolution in RIXS could explain the observed low damping. Below, we will show in reply to question #5 of the reviewer that with the available experimental resolution in RIXS damping factors γ above 5 meV already lead to a clear, experimentally observable broadening of the magnon peak. Here we would like to point out why we think that INS will likely not be able to provide with better data. We agree with the referee that for low energy magnetic excitations up to tens of meV, INS is undoubtedly the means of choice because of its very high energy resolution, far surpassing the one achievable with RIXS. However, for energy losses well above 100 meV INS becomes a much more challenging experiment in terms of resolving power, count rates and signal to background ratio and becomes comparable in performance to RIXS. The visual comparison of the INS results for CaFe₂As₂ that the reviewer mentioned and our RIXS raw data without any background subtraction in the figure below gives clear evidence of that. Another particular advantage of RIXS for this study is the resonant character

of the technique, which allows us to be sure that the detected excitations originate from Co 3d states. Thirdly, an INS study of the magnetic excitations in CeCo_2P_2 is also impeded by the very large absorption and very weak coherent scattering cross section of Co compared to for instance Fe, Ni, or Cu (see Table R1). We therefore think that, in this particular case, INS will likely not be able to provide qualitatively better data than RIXS.

	σ_{coh} (barn)	σ_{incoh} (barn)	σ_{abs} (barn)
Fe	11.22	0.4	2.56
Co	0.78	4.8	37.18
Ni	13.3	5.2	4.59
Cu	7.48	0.55	3.78

V. F. Sears: Neutron scattering lengths and cross section. Neutron News 3, 26 (1992)

Table R1: Coherent vs. incoherent scattering cross section and the absorption cross section of Co for thermal neutrons, compared to those of the neighbouring 3d elements.

Figure R2: INS data for the high-energy magnon excitations in NaFeAs , SrFe_2As_2 and CeFe_2As_2 , compared with our RIXS data on CeCo_2P_2 in the same energy range. Clearly visible is the comparable resolution between RIXS and INS.

Secondly, the reviewer ask about the distinction between strongly and weakly or undamped spin wave systems. The main criterion to consider is the damping ratio $2\gamma/\hbar\omega_q$. The smaller the damping ratio the longer the magnon life times, and in weakly damped systems, $2\gamma/\hbar\omega_q \ll 1$. We agree with the referee that $\gamma = 0$ will never be achieved in any magnetically ordered system because of, for instance, spin wave scattering off domain walls or thermally populated low energy spin waves or other quasi-particles. However, in metals the dominant term to spin wave life times is thought to be Landau damping, i.e. spin wave scattering off the Stoner continuum. We have used the term “undamped spin waves in a metal” in our paper because both our experiment as well as our calculations show that the classical damping mechanism in metals, Landau damping, is suppressed in this system and the observed spin wave life times are comparable to those of magnetic insulators. This is well seen in comparison with other metallic systems. For example, FePd thin films with a ratio $\gamma/\hbar\omega_q \sim 1/6$ and magnon energies in excess of 100 meV are considered as metals hosting long lived, i.e. weakly damped, magnons (“Long-living terahertz magnons in ultrathin metallic ferromagnets”, Nature Comm. 6, 6126 (2015)). In our case, the spin wave damping is several times smaller and the spin waves reach energies twice as large. We also note that CaFe_2As_2 evidences much weaker damping compared to, for instance, its Ba counterpart, although its damping factor is still several times that of CeCo_2P_2 . We have added CaFe_2As_2 as a reference

to our manuscript and used it, together with other examples of metallic magnets, to generalise the discussion of our findings at the end of the manuscript (see also new Fig. 4). Furthermore, we changed the title of our manuscript from “Undamped spin waves in an itinerant antiferromagnet” to “Long-lived spin waves in a metallic antiferromagnet” in order to avoid any confusion.

Figure R3: New Figures 4b,c where we compared the magnon damping in CeCo_2P_2 with that found in other metallic magnets, incl. CaFe_2As_2 . Note the logarithmic scale.

Reviewer 3:

1. As authors mentioned that their CeCo_2P_2 is a metallic antiferromagnet. Are they able to provide more experimental evidences to demonstrate that it exhibits metallic properties.

Authors:

We have added a section in the Supplementary Information showing the results of our resistivity measurements on both compounds, CeCo_2P_2 and LaCo_2P_2 . We have taken great care to determine the absolute values of the resistivity as precise as possible. To this end, we have used a very thin single crystal (40 μm thick) with nearly perfect quadratic shape (inset of the figure below). Using a van-der-Pauw analysis [L. J. van der Pauw, Philips Res. Repts. 13, p. 1 (1958)], we could determine reliable absolute values of the resistivity. From the temperature dependence of the resistivity and a room temperature value of about 260 $\mu\text{Ohm cm}$ we can clearly demonstrate that CeCo_2P_2 is a metal. Around 50 K, we see a smooth kink in the resistivity data. This feature is present in measurements on different samples, although its precise temperature varies from sample to sample. Presently, we do not know the origin of this change in resistivity and need to analyse this further. However, for our present claim of metallicity this is not important.

Figure R4: Resistivity as function of temperature for single crystals of CeCo_2P_2 (blue symbols) and LaCo_2P_2 (black symbols). For CeCo_2P_2 we used a van-der-Pauw method to extract absolute values of the resistivity. The CeCo_2P_2 crystal with the four contacts made with silver paste and Pt wires is shown in the inset.

Reviewer 3:

2. Authors mentioned “This very intense fluorescence feature due to local, intraatomic decay of the RIXS intermediate state is a characteristic of RIXS spectra taken from metallic systems with high electron mobility” in the page 8. I'm interested in finding out whether this can differentiate the surface state of CeCo_2P_2 , given that there is a theoretical computation suggesting that it is a topological semi-metal.

Authors:

While RIXS has enough sensitivity to measure down to single layers, those samples need to be specifically prepared as single layers. For a bulk sample like in our case the technique is not sensitive to surface effects.

Reviewer 3:

3. The authors referred to CeCo_2P_2 as an itinerant antiferromagnet in the title. Whether authors try to state the origin of magnetism in CeCo_2P_2 arises from quasiparticle excitations of a nested Fermi surface? If not, this title may be misleading to readers.

Authors:

We named CeCo_2P_2 an itinerant antiferromagnet, meaning an itinerant electron or metallic antiferromagnet. This wider definition of itinerant magnetism that does not explicitly discuss nesting properties of the Fermi surface is quite commonly used (e.g. “Two-dimensional itinerant ferromagnetism in atomically thin Fe_3GeTe_2 ” Nature Materials 17, 778 (2018)”). Fedders and Martin in their paper “Itinerant Antiferromagnetism” [Phys. Rev. 143, 245 (1966)] used the term in the narrower sense that the referee refers. While the metallic character of CeCo_2P_2 is clearly established, the nesting properties of the Fermi surface and possible interband quasiparticle excitations across it have not been determined yet (to the best of our knowledge). We agree with the referee, that in order to avoid

confusion it is best to replace the term itinerant antiferromagnet by metallic antiferromagnet in our manuscript, which we have done in the revised version.

Reviewer 3:

4. Have temperature-dependent measurements been conducted to verify that the inelastic peaks in the RIXS spectra are magnetic excitations?

Authors:

Temperature dependent measurements were conducted at $\mathbf{q} = (0.24 \ 0 \ 1)$. The results are shown in Figure R5 below and have been included as a Supplementary Information now. The intensity and stiffness of the magnon excitations slightly decrease towards room temperature, as expected, but the effect is small due to the very high ordering temperature. Within the statistical errorbars we do not see a significant evolution of the damping factor γ as a function of temperature, in agreement with previous findings [Phys. Rev. Lett. 118, 127203 (2017)].

Figure R5: (a) Measured RIXS spectra and the results of their fits for different temperatures. The spectra are vertically offset for visualization. The magnon peak convolved by the experimental resolution is highlighted as filled curve. (b) Temperature dependence of the intensity and excitation energy $\hbar\omega_{\mathbf{q}}$ of the magnon peak.

Reviewer 3:

5. I attempted to extract the damping factor from the RIXS spectra of CeCo2P2 using the function described in the main text. However, the value I obtained was approximately 14 meV at 60 meV, which is almost twice the value shown in the data plot in Fig. 4b. To enable readers to replicate the analysis, the authors should provide a more explicit explanation of how they extracted the damping factor.

Authors:

As mentioned in the manuscript the experimental raw data were fit to a simple model containing one line at zero energy to account for quasielastic scattering and one peak to model the magnon excitation with the widely used expression (1) for a damped harmonic oscillator model.

$$I(E_{\text{loss}} > 0, \hbar\omega_q, \gamma) = I_0(\hbar\omega_q) \frac{\gamma E_{\text{loss}}}{(E_{\text{loss}}^2 - \hbar^2\omega_q^2)^2 + 4\gamma^2 E_{\text{loss}}^2} \quad (1)$$

Both the zero energy line and the damped harmonic oscillator were convolved with the experimentally determined energy resolution. This yields a Gaussian peak at zero energy with energy resolution limited FWHM and a magnon peak broadened by the experimental resolution. The structureless, weak background of the high energy fluorescence peak was approximated by a second order polynomial. The statistical errorbars on each point in the spectrum were included in the fit analysis. The errorbars in gamma were determined from the resulting covariance matrix of the fit.

Below we show the results of this fit analysis for different q values including the one analysed by the referee. We also plot the curves that one would expect for different values of the damping factor gamma. It can be seen that γ values of 10 meV already lead to notably wider peaks than experimentally observed. Clearly a γ value of about 14 meV as found by the referee cannot describe the experimental data. We have added a section to the Supplementary Information where the fitting procedure is explained in more detail. We also included the above figure so that the readers can convince themselves that γ values well below 10 meV can be easily extracted from the RIXS data.

Figure R6: Top panel: RIXS raw data taken at different q together with the results of our fit analysis. The expected line shapes of the magnon excitations for $\gamma = 14$ meV and $\Delta E = 28$ meV are shown as fine dashed lines. Bottom panel: The same data after subtraction of a Gaussian elastic line with FWHM = 28 meV and the background. The data are compared with the expected line shape for $\Delta E = 28$ meV and increasing γ values.

We would like to point out that the widely established expression (1) yields a peak with FWHM 2γ , not γ , for the damped spin wave excitation. This might explain the factor two that the referee found, as often the damping factor γ is also defined as the FWHM, and not the HWHM of the magnon excitation. In this case the above expression (1) writes as

$$I \propto \frac{\gamma E_{\text{loss}}}{(E_{\text{loss}}^2 - \hbar^2 \omega^2)^2 + \gamma^2 E_{\text{loss}}}$$

We have carefully rechecked which definition of γ was used in each publication that we compare our data with in Fig. 4b and the new Fig. 4c to make sure that all data are plotted with the same convention for the damping factor γ . We found that in some cases, e.g. FePd thin film, we compared to the reported FWHM. We also realised that the results of our calculations were reported as 2γ (FWHM) in the previous Fig. 4b, and not γ . Both has been corrected in the revised version. The conclusions of our paper are unaffected. We thank the referee for this very helpful comment.

Reviewer 3:

6. The authors should take great care to review the manuscript for spelling errors. Mistakes such as "overestimateded", "calculated", "interant", and so on should be avoided.

Authors:

We have gone through the manuscript again and carefully checked for spelling mistakes.

We highly appreciate both reviewers for the time they spent in reading and evaluating our manuscript, and the constructive suggestions, remarks and questions. We believe that they are well accounted for in the revised manuscript and with the additional information provided in the Supplementary Information File.

References:

- [R1] Reehuis *et al.* Journal of Alloys and Compounds **266**, 54 (1998).
- [R2] Kliemt *et al.* Cryst. Res. Technol. **55**, 1900116 (2020).
- [R3] Reehuis *et al.* Phys. Chem. Solids **51**, 961 (1990).
- [R4] Tian *et al.* Physica B **512**, 75 (2017).
- [R5] Poelchen *et al.* ACS Nano **16**, 3573 (2022).

Reviewers' Comments:

Reviewer #1:

Remarks to the Author:

I have checked the revised manuscript and the response from the authors. I think my concerns have been convincingly addressed and reflected in the revised manuscript. I have no further questions.

Reviewer #3:

Remarks to the Author:

The authors have effectively addressed all of my concerns and have made significant improvements to the manuscript. They have discovered a unique metallic antiferromagnet, CeCo₂P₂, which exhibits long-lived spin waves and holds importance for practical applications. Based on the revisions that have been made, I believe that the paper deserves publication in Nature Communications.

I appreciate the authors' decision to change the title. While the term "itinerant ferromagnet" is quite commonly used, the term "itinerant antiferromagnet" would imply the involvement of nesting properties of the Fermi surface. The previous classification of CeCo₂P₂ could lead to confusion among readers in the field of condensed matter physics. I have one suggestion that the authors should consider before publication. It is possible that a larger Stoner gap, as depicted in Fig.1b, could be responsible for the long-lived spin waves observed. However, an alternative explanation could be a narrow localized Stoner continuum, which can arise from quasiparticle excitations between spin-up and spin-down flat bands (as mentioned in Communications Physics 4, 240 (2021)). Experimental observations using ARPES have revealed the presence of flat bands in CeCo₂P₂. Therefore, it is plausible that the spin-up and spin-down flat bands contribute to the phenomenon in question. Fortunately, the authors have presented resistivity results to support their arguments, but the influence of flat bands has not been explicitly ruled out. Hence, it would be beneficial for the authors to provide a discussion regarding the different origins of the long-lived spin waves in CeCo₂P₂, which would enhance the overall physical significance of the paper.

We would like to thank the reviewers again for the effort that they put into reviewing our manuscript. Their comments helped us to improve our manuscript substantially.

Reply to Reviewer #1

Reviewer #1:

I have checked the revised manuscript and the response from the authors. I think my concerns have been convincingly addressed and reflected in the revised manuscript. I have no further questions.

Authors:

We are happy about the positive evaluation by Reviewer #1.

Reply to Reviewer #3

Reviewer #3:

The authors have effectively addressed all of my concerns and have made significant improvements to the manuscript. They have discovered a unique metallic antiferromagnet, CeCo₂P₂, which exhibits long-lived spin waves and holds importance for practical applications. Based on the revisions that have been made, I believe that the paper deserves publication in Nature Communications.

Authors:

We are happy about the positive evaluation by Reviewer #3.

Reviewer #3:

I appreciate the authors' decision to change the title. While the term "itinerant ferromagnet" is quite commonly used, the term "itinerant antiferromagnet" would imply the involvement of nesting properties of the Fermi surface. The previous classification of CeCo₂P₂ could lead to confusion among readers in the field of condensed matter physics. I have one suggestion that the authors should consider before publication. It is possible that a larger Stoner gap, as depicted in Fig.1b, could be responsible for the long-lived spin waves observed. However, an alternative explanation could be a narrow localized Stoner continuum, which can arise from quasiparticle excitations between spin-up and spin-down flat bands (as mentioned in Communications Physics 4, 240 (2021)). Experimental observations using ARPES have revealed the presence of flat bands in CeCo₂P₂. Therefore, it is plausible that the spin-up and spin-down flat bands contribute to the phenomenon in question. Fortunately, the authors have presented resistivity results to support their arguments, but the influence of flat bands has not been explicitly ruled out. Hence, it would be beneficial for the authors to provide a discussion regarding the different origins of the long-lived spin waves in CeCo₂P₂, which would enhance the overall physical significance of the paper.

Authors:

We are thankful to Reviewer #3 for bringing up the point of localized states and their influence on electron-magnon interaction. Based on the Reviewer's suggestion, we included the following statement in the manuscript (lines 290 - 299):

"We note that recently the effect of a narrow Stoner continuum on spin wave excitations has been discussed [Y. Xie et al., *Communications Physics* 4, 240 (2021)]. Such a narrow Stoner continuum would arise from very flat, spin polarised bands around E_F . In such a scenario Stoner spin flip excitations would occur only in a very narrow energy band and long lived spin-waves could possibly exist at energies above and below. Our electronic structure calculations (Figure 1d) as well as recent ARPES data [G. Poelchen et al., *ACS Nano* 2022, 16, 3, 3573–3581] give no indication of flat bands in the bulk of CeCo₂P₂. The very low Sommerfeld coefficient found in specific heat measurements (Supplementary Information, section 1) further excludes the presence of heavy quasiparticles close to E_F . Therefore, we believe that a scenario based on a very localised Stoner continuum cannot be invoked for CeCo₂P₂."